# Loss of MeCP2 disrupts cell autonomous and autocrine BDNF signaling in mouse glutamatergic neurons

Charanya Sampathkumar[1,2], Yuan-Ju Wu[1,2], Mayur Vadhvani[2,3], Thorsten Trimbuch[1,2], Britta Eickholt[2,3], Christian Rosenmund[1,2]*

[1]Department of Neurophysiology, Charité Universitätsmedizin Berlin, Berlin, Germany; [2]NeuroCure Cluster of Excellence, Charité Universitätsmedizin Berlin, Berlin, Germany; [3]Institute of Biochemistry, Charité Universitätsmedizin Berlin, Berlin, Germany

**Abstract** Mutations in the *MECP2* gene cause the neurodevelopmental disorder Rett syndrome (RTT). Previous studies have shown that altered MeCP2 levels result in aberrant neurite outgrowth and glutamatergic synapse formation. However, causal molecular mechanisms are not well understood since MeCP2 is known to regulate transcription of a wide range of target genes. Here, we describe a key role for a constitutive BDNF feed forward signaling pathway in regulating synaptic response, general growth and differentiation of glutamatergic neurons. Chronic block of TrkB receptors mimics the MeCP2 deficiency in wildtype glutamatergic neurons, while re-expression of BDNF quantitatively rescues MeCP2 deficiency. We show that BDNF acts cell autonomous and autocrine, as wildtype neurons are not capable of rescuing growth deficits in neighboring MeCP2 deficient neurons *in vitro* and *in vivo*. These findings are relevant for understanding RTT pathophysiology, wherein wildtype and mutant neurons are intermixed throughout the nervous system.

*For correspondence: christian. rosenmund@charite.de

## Introduction

Rett syndrome (RTT) is a severe progressive neurodevelopmental disorder, mainly caused by mutations in the X-linked gene encoding methyl-CpG binding protein 2 (MeCP2), a protein involved in transcriptional regulation (*Amir et al., 1999*; *Wan et al., 1999*; *Xiang et al., 2000*). RTT patients show regression of head growth followed by various neurological symptoms including seizures, mental retardation, stereotypic hand-wringing movements, breathing irregularities, ataxia and autistic behavior (*Rett, 1966*). Mouse models with *Mecp2* mutations display similar neurological phenotypes, and have been quite critical in defining the pathophysiology of RTT. *Mecp2^Null/y* mice grow normally until 4–6 weeks of age, after which they display RTT-like symptoms such as reduced mobility, hindlimb clasping, abnormal breathing patterns and premature death (*Chen et al., 2001*; *Guy et al., 2001*). Similarly, mice engineered to express twice the endogenous levels of MeCP2 (*Mecp2^Tg1*) are characterized by seizures, forepaw clasping, hypoactivity, increased aggression, and around 30% die by one year of age (*Collins et al., 2004*; *Jugloff et al., 2008*; *Luikenhuis et al., 2004*). Loss or doubling of MeCP2 in primary mouse hippocampal neurons results in reduction or enhancement of synaptic response respectively, primarily due to the number of glutamatergic synapses formed (*Chao et al., 2007*). Furthermore, restoring MeCP2 levels rescued neurological defects associated with loss of MeCP2 the extent of which depends on timing of activation and dynamic variation in MeCP2 levels (*Giacometti et al., 2007*; *Guy et al., 2007*). Besides MeCP2, manipulation of target genes regulated by MeCP2 by genetic means or by indirect and nonspecific mechanisms has

**eLife digest** Rett syndrome is a progressive brain disorder. Individuals with the condition (who are typically girls) grow normally until they are 6-18 months old and then developmentally regress, with symptoms including anxiety, impaired coordination, seizures and breathing problems.

Rett syndrome is caused by mutations in the gene that encodes a protein called MeCP2. Researchers know that MeCP2 is vital for "excitatory" neurons in the brain to communicate with (and activate) their neighbors. Neurons that lack MeCP2 tend to make fewer of the connections across which they communicate – called synapses – with others.

Many researchers who study Rett syndrome use male mice that lack the MeCP2 protein. This mouse model mimics the symptoms seen in Rett patients, but at a faster and more severe rate. These studies have shown that restoring normal levels of the protein in neurons prevents the majority of Rett-like symptoms in these mice and reverses the disorder.

MeCP2 controls the activity of a number of other genes. These include the gene that produces a protein called Brain-Derived Neurotrophic Factor (BDNF), which helps neurons to grow. Sampathkumar et al. have now studied neurons from mouse models of Rett syndrome to investigate whether BDNF can overcome the defects seen in neurons that lack MeCP2. Viewed under a high-powered microscope, the Rett-like neurons appear smaller than healthy neurons and form fewer synapses. However, increasing the amount of BDNF in the diseased neurons restores normal growth and enables the cells to form more synapses.

Girls with Rett syndrome tend to have a mixture of healthy neurons and those that do not produce the right amount of MeCP2. To mimic this, Sampathkumar et al. grew a mixture of normal and Rett-like mouse neurons in a culture dish. The healthy neurons did not help the diseased neurons to form the correct number of synapses. However, increasing the levels of BDNF in the Rett-like neurons enhanced their ability to form synapses, and increased their cell size to match their healthy counterparts.

Further work is now required to uncover whether manipulating the gene that encodes BDNF – or other genes that MeCP2 controls the activity of – in the brain can reduce the symptoms and slow the progression of Rett syndrome.

resulted in alleviating various features of RTT (*Deogracias et al., 2012*; *Johnson et al., 2012*; *Kondo et al., 2008*; *Kron et al., 2014*; *Ogier et al., 2007*; *Schmid et al., 2012*; *Tropea et al., 2009*).

Functional interaction between MeCP2 and brain-derived neurotrophic factor (BDNF) has been reported in various studies and MeCP2 has been known to regulate expression of BDNF (*Chahrour et al., 2008*; *Chang et al., 2006*; *Martinowich et al., 2003*). *Bdnf* deletion from postnatal forebrain excitatory neurons of MeCP2 mutant mice resulted in an earlier onset of RTT while conditional BDNF overexpression delayed RTT onset and normalized spontaneous activity (*Chang et al., 2006*). Similarly, endogenous MeCP2 knockdown reduced dendritic length in E18 hippocampal neurons, which was fully rescued upon BDNF overexpression (*Larimore et al., 2009*). Considering that BDNF is an important regulator of neurite outgrowth and synapse formation (*Cheng et al., 2011*; *Finsterwald et al., 2010*; *Gottmann et al., 2009*; *McAllister et al., 1999*; *Park and Poo, 2013*; *Poo, 2001*; *Tolwani et al., 2002*; *Wang et al., 2015*), it is essential to understand effects of BDNF modulation in $Mecp2^{Null/y}$ neurons specifically in steering neuronal growth and glutamatergic synapses. Early work has focused mainly on phenotypic rescue upon increasing BDNF levels in MeCP2-deficient mice but mechanistic interactions have not been resolved.

In this study, we examine putative causal mechanisms of BDNF impacting cellular growth as well as the formation of excitatory synapses in hippocampal neurons lacking MeCP2. We found that increased BDNF levels restored synaptic output and morphological phenotypes in MeCP2 deficient neurons in a cell autonomous and autocrine manner *in vitro* and *in vivo*. Importantly, blocking the BDNF pathway converted wildtype neurons to a phenotype that mimicked MeCP2-deficiency induced defects. These findings add to the current picture of BDNF signaling in stabilizing various

features of RTT patients and are crucial for the profound understanding of BDNF-mediated therapeutic strategies and more generally, pathophysiology of the RTT disease.

## Results

### Loss, but not doubling of MeCP2, alters neurite outgrowth and neuronal soma size in hippocampal glutamatergic neurons

We previously reported that loss or doubling of MeCP2 results in altered glutamatergic synapse number in vitro and in vivo (*Chao et al., 2007*) underlining tightly controlled MeCP2 dosage as a prerequisite for optimal synapse formation. Why MeCP2 levels are rate limiting in glutamatergic synapse formation is not clear, and could include pre- or postsynaptic factors that regulate synapse formation as well as factors that regulate general neuronal development. Hence, we began this analysis by examining the role of neurite outgrowth. Single-cell hippocampal autaptic cultures were utilized and particularly beneficial due to the unambiguous assignment of axon and dendrite and the ability to restrict analysis to a cell autonomous condition.

Axonal and dendritic outgrowths were measured during the first 12 days in vitro (DIV) from *Mecp2$^{Tg1}$* and *Mecp2$^{Null/y}$* neurons as well as control neurons derived from their respective WT littermates (*Figure 1*). Neurons were labeled with Tau1 and MAP2 to mark for axons and dendrites respectively (*Figure 1A*). Comparative analysis revealed that *Mecp2$^{Tg1}$* neurons appeared normal and did not display any morphological deficits, with axonal and dendritic outgrowth similar to WT neurons (*Figure 1C and E*). However, *Mecp2$^{Null/y}$* neurons showed a significant reduction of 23–41% and 27–41% in overall length of axons and dendrites respectively (*Figure 1B and D*). Considering these deficits in general growth and differentiation in *Mecp2$^{Null/y}$* neurons, we analyzed the size of neuronal somata and observed a 24–33% reduction in soma size while *Mecp2$^{Tg1}$* neuronal soma remained unchanged (*Figure 1F–H*). These results do not explain the Tg1 gain-of-function

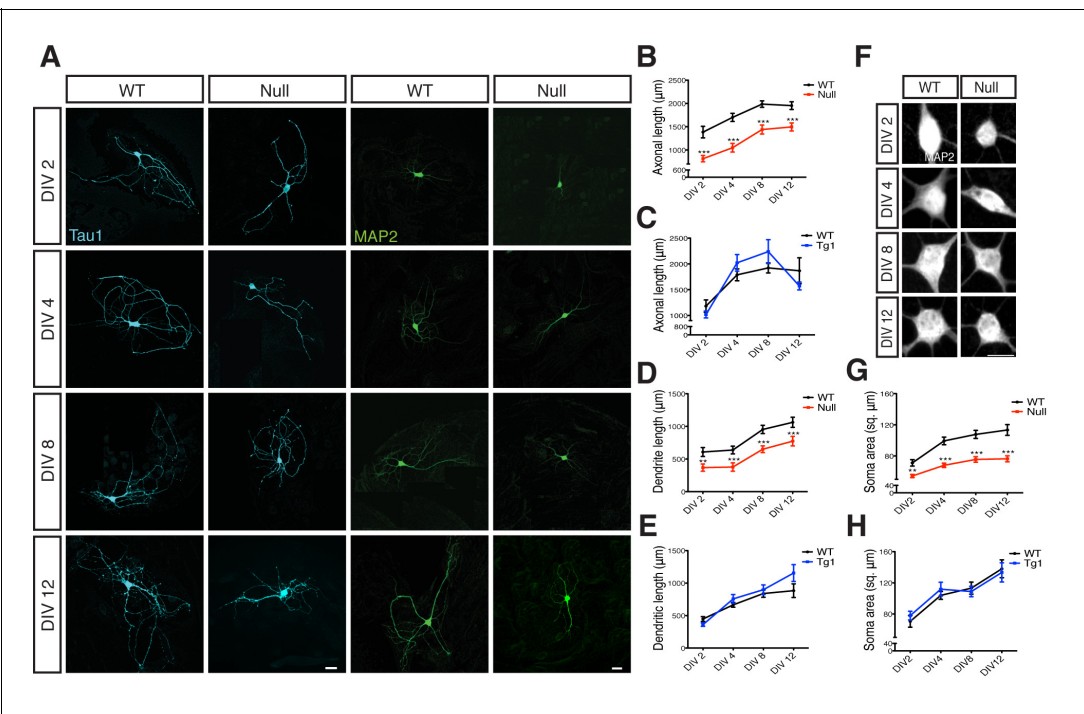

**Figure 1.** Loss of MeCP2 alters neurite outgrowth and neuronal soma size. (A) Co-immunostaining of glutamatergic hippocampal neurons from WT and *Mecp2$^{Null/y}$* mice at DIV 2, 4, 8 and 12 for Tau1 (cyan) and MAP2 (green). Scale bars represent 20 µm. (B–E) Mean axonal length and dendritic length measured at different time points for *Mecp2$^{Null/y}$* (B,D) and *Mecp2$^{Tg1}$* neurons (C,E). (F) Somatic labeling of glutamatergic hippocampal neurons from WT and *Mecp2$^{Null/y}$* mice at DIV 2, 4, 8 and 12 with MAP2. Scale bar represents 20 µm. (G,H) Mean soma area measured at different time points for *Mecp2$^{Null/y}$* (G) and *Mecp2$^{Tg1}$* neurons (H). Data shown as mean ± SEM. **p<0.01; ***p<0.001.

phenotype suggesting that an alternate mechanism may be necessary for enhanced synapse number. Additionally, the morphological findings reveal a general growth deficiency in glutamatergic neurons lacking MeCP2 possibly accounting for decreased synapse number.

## BDNF overexpression in *Mecp2^Null/y* autaptic neurons restores evoked EPSC magnitude, RRP size, synapse number, dendrite length, soma- and nucleus size

MeCP2 is a transcriptional regulator that activates or represses expression of several downstream genes based on cell type, origin, age and heterogeneity of brain region (*Chahrour et al., 2008*; *Tudor et al., 2002*). Given that BDNF (i) is a consistent neuronal target gene of MeCP2 (*Chahrour et al., 2008*; *Martinowich et al., 2003*), (ii) is an essential regulator of synapse formation and dendritic complexity (*Finsterwald et al., 2010*; *McAllister et al., 1999*; *Tolwani et al., 2002*), and that (iii) levels are reduced in MeCP2 knockout mice (*Chang et al., 2006*); we investigated the mechanistic role of BDNF signaling in regulating synaptic output and synapse formation in *Mecp2-Null/y* neurons at single-cell level.

To address this question, we utilized the lentiviral system driven by a synapsin promoter to over-express BDNF in *Mecp2^Null/y* and WT neurons to attempt rescue of growth deficits (*Figure 2A*). WT and *Mecp2^Null/y* neurons expressing GFP were used as control. Indeed, BDNF overexpression in *Mecp2^Null/y* neurons fully restored all morphological parameters analyzed. Particularly, glutamatergic synapse number, dendrite length and soma size in *Mecp2^Null/y*neurons were reduced down to 51, 58 and 72% of WT and rescued up to 105, 89 and 102% of that of WT glutamatergic neurons respectively upon BDNF overexpression (*Figure 2B–E*). Similarly, nucleus size was reduced by 26% in *Mecp2^Null/y*neurons and rescued up to 95% in BDNF overexpressing *Mecp2^Null/y* neurons (*Figure 2—figure supplement 1*). Overexpressing BDNF in WT neurons itself did not impact synapse number, dendrite length or soma size (*Figure 2L*). Further, lack of enhancement in synapse number upon BDNF overexpression in WT neurons verified that the *Mecp2^Tg1* gain-of-function phenotype was not related to BDNF function and ascribed to potentially separate mechanism(s).

We next examined functional implications of the observed morphological defects by measuring synaptic output in *Mecp2^Null/y* neurons overexpressing BDNF. *Mecp2^Null/y* neurons revealed 54% decrease in evoked EPSC amplitude (2.92 ± 0.42 nA, p<0.05) and 58% decrease in RRP charge (0.37 ± 0.05 nC, p<0.01) as compared to WT neurons (EPSC: 6.38 ± 0.9 nA; RRP: 0.89 ± 0.14 nC), both of which were rescued back up to 113 and 87% of WT levels, respectively (EPSC: 7.23 ± 0.97 nA, p<0.01; RRP: 0.77 ± 0.09 nC, p<0.01) (*Figure 2F–I*). We did not observe any differences in short-term plasticity or release efficiency in either WT or *Mecp2^Null/y* neurons with or without BDNF overexpression, as reflected by the paired-pulse ratio (PPR) and vesicular release probability ($P_{vr}$) measurements (*Figure 2J–L*). These findings further illustrate that interaction of MeCP2 and BDNF was predominantly related to scaling of neuronal size, growth and synaptic response.

## Both TrkB inactivation and BDNF neutralization revert synaptic output and synapse number back to *Mecp2^Null/y* levels

Overall, synaptic and morphological deficits resulting from loss of MeCP2 function were fully re-established by restoring the ability of *Mecp2^Null/y* neurons to synthesize increased levels of BDNF. We now asked if disruption of BDNF binding to TrkB could negate effects of BDNF overexpression in *Mecp2^Null/y* neurons. The low molecular weight TrkB ligand, ANA-12, selectively binds to the TrkB receptor thereby blocking BDNF-induced TrkB activation and inhibiting intracellular signaling cascades downstream of TrkB (*Cazorla et al., 2011*). Application of ANA-12 to the BDNF overexpressing *Mecp2^Null/y* neurons reverted both synaptic output (evoked EPSC amplitude: 2.83 ± 0.47 nA, p<0.001 and RRP size: 0.19 ± 0.02 nC, p<0.001) (*Figure 2G and I*) and morphological phenotypes (synapse number: 151 ± 10, p<0.01; dendrite length: 748 ± 46 μm, p<0.05; soma size: 93 ± 4 μm², p<0.05 and nucleus size: 45 ± 2 μm², p<0.01) (*Figure 2C–E* and *Figure 2—figure supplement 1*) back to *Mecp2^Null/y* control levels. Control groups included WT and *Mecp2^Null/y* neurons both with and without application of ANA-12. Interestingly, *Mecp2^Null/y* neurons treated with ANA-12 did not undergo any further loss-of-function effect and remained comparable to *Mecp2^Null/y* control neurons (*Figure 2G and I*) indicating that the BDNF-TrkB canonical pathway was quantitatively disrupted in MeCP2-deficient neurons. On the other hand, WT neurons treated with ANA-12 displayed

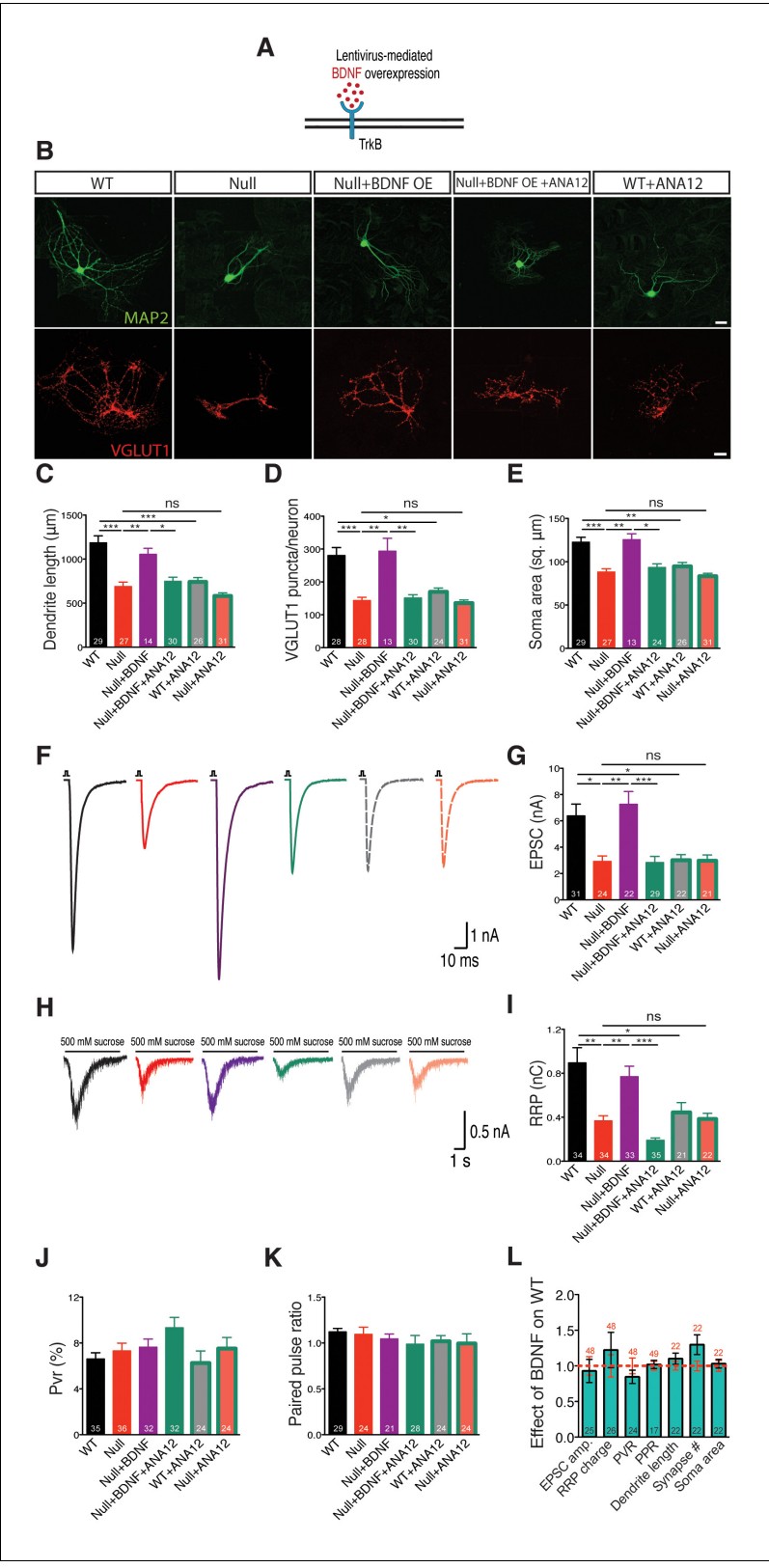

**Figure 2.** BDNF overexpression in *Mecp2^{Null/y}* autaptic neurons normalizes cell morphology and restores synaptic output. (**A**) Experimental scheme of lentivirus-mediated BDNF overexpression in *Mecp2^{Null/y}* neurons, showing neuronal membrane (black), BDNF (red) and TrkB receptor (blue). (**B**) Representative images of neuronal morphology under the following conditions (from left to right): WT, Null, Null + BDNF, Null + BDNF + ANA12, WT

*Figure 2 continued on next page*

*Figure 2 continued*

+ ANA12. Dendritic outgrowth (top) and glutamatergic synapses (bottom) indicated by MAP2 (green) and VGLUT1 (red) labeling. Scale bars represent 20 μm. (C–E) Bar graphs show mean dendrite length (C), glutamatergic synapse number (D) and neuronal soma area (E). (F) Representative traces of evoked EPSCs recorded from autapses under the following conditions: WT, Null, Null + BDNF, Null + BDNF + ANA12, WT + ANA12, Null + ANA12. (G) Bar graph shows mean evoked EPSC amplitude. (H) Representative traces of average postsynaptic response to 5 s application of 500 mM sucrose. Experimental groups same as (F). (J–K) Bar graphs show mean RRP size (I), vesicular release probability $P_{vr}$ (J) and paired-pulse ratio with 25 ms inter-stimulus interval (K). (L) Bar graph shows EPSC amplitude, RRP size and $P_{vr}$, PPR and dendrite length, glutamatergic synapse number and soma area, normalized to WT (dashed red line). Number of neurons (n) shown in the bars for all graphs. Data shown as mean ± SEM. *p<0.05; **p<0.01; ***p<0.001; ns: not significant.

The following figure supplement is available for figure 2:

**Figure supplement 1.** BDNF overexpression in *Mecp2^{Null/y}* autaptic neurons normalizes nuclei size.

morphological as well as synaptic deficits and behaved similar to *Mecp2^{Null/y}* neurons (*Figure 2G and I*). As before, PPR and $P_{vr}$ remained unchanged in all conditions (*Figure 2J–L*). These results strongly suggest that BDNF synthesis and an active BDNF-TrkB pathway are essential for normal neuronal growth in WT as well as RTT-like hippocampal glutamatergic neurons.

Consistent with these results, we show that treatment of BDNF overexpressing *Mecp2^{Null/y}* neurons with an anti-BDNF neutralizing antibody (*Figure 3A*) decreased EPSC amplitude and RRP size by 50 and 53% respectively (*Figure 3B–E*), thereby negating phenotype rescue seen via BDNF overexpression. Thus, we reveal that BDNF synthesis was clearly impaired in glutamatergic neurons lacking MeCP2. To further test the specificity of phenotype rescue, an alternate neurotrophic factor, nerve growth factor (NGF), was overexpressed in WT and *Mecp2^{Null/y}* neurons (*Figure 3—figure supplement 1A*). We found that NGF failed to restore normal synaptic transmission, emphasizing the specificity of the role of BDNF-TrkB in *Mecp2^{Null/y}* neurons (*Figure 3—figure supplement 1B and C*). PPR and $P_{vr}$ measurements remained unchanged across all conditions in both BDNF neutralization (*Figure 3F and G*) and NGF overexpression experiments (*Figure 3—figure supplement 1D and E*). These findings show that abolishing BDNF function as well as BDNF-induced TrkB activation affected synapse formation in *Mecp2^{Null/y}* autaptic neurons, postulating the mechanistic role of BDNF in regulating synaptic function in RTT-like excitatory neurons.

## Exogenous BDNF application in *Mecp2^{Null/y}* neurons rescues physiological and morphological phenotypes

If BDNF synthesis were indeed disrupted and led to general growth and synaptic deficits in *Mecp2^{Null/y}* neurons, exogenous application of BDNF should be able to bypass this deficiency and rescue loss-of-function. Neurons were treated with 50 ng/ml of recombinant human BDNF at DIV 2 and replenished every 2–3 days until DIV 14 when they were either taken for EPSC and RRP measurements or fixed for morphology analysis (*Figure 3H*).

Exogenous BDNF application normalized evoked EPSC amplitude (*Figure 3I*) and RRP size (*Figure 3J*) as well as re-established normal dendritic outgrowth (*Figure 3K*) and glutamatergic synapse number (*Figure 3L*) in *Mecp2^{Null/y}* neurons. All restored phenotypes were specific to BDNF-TrkB since application of ANA-12 to *Mecp2^{Null/y}* neurons treated with exogenous BDNF reverted evoked EPSC amplitude (44% decrease) and RRP size (66% decrease) as well as synapse number (39% decrease), dendrite length (47% decrease) and soma size (28% decrease) back to *Mecp2^{Null/y}* levels (*Figure 3I–M*). No significant changes were seen either in PPR or $P_{vr}$ measurements upon BDNF application (*Figure 3—figure supplement 2A and B*). These findings confirmed that *Mecp2^{Null/y}* neurons were deficient in BDNF synthesis and revealed disrupted BDNF-TrkB signaling, which was bypassed upon exogenous application of BDNF.

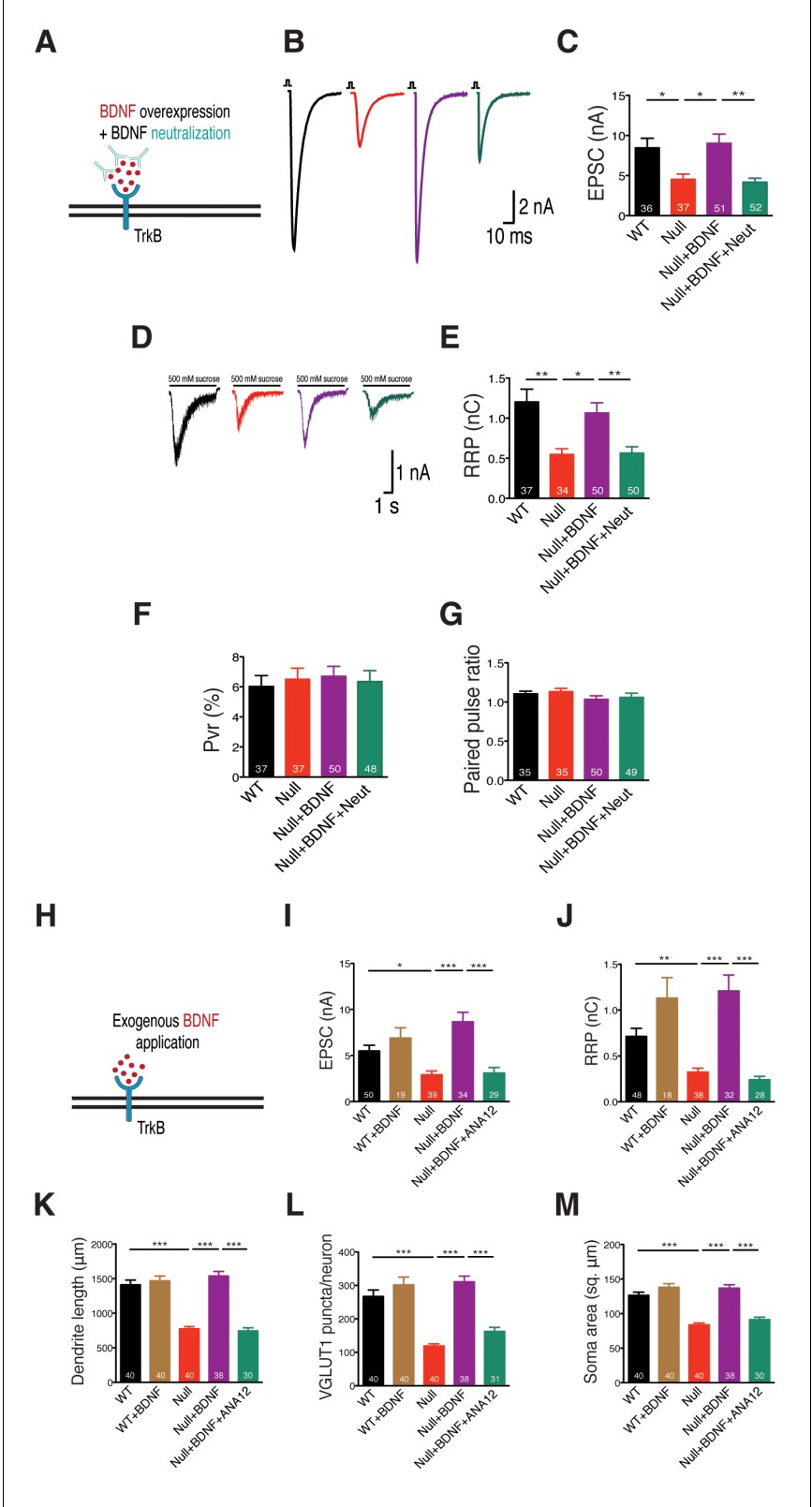

**Figure 3.** BDNF neutralization fails to rescue physiological phenotypes while exogenous application of BDNF restores physiological and morphological phenotypes, reaffirming specificity of BDNF-TrkB interaction in *Mecp2^{Null/y}* glutamatergic neurons. (**A**) Experimental scheme of lentivirus-mediated BDNF overexpression and BDNF neutralization in *Mecp2^{Null/y}* neurons, showing neuronal membrane (black), BDNF (red), TrkB receptor (blue)

*Figure 3 continued on next page*

*Figure 3 continued*

and BDNF neutralizing antibody (cyan). (**B**) Representative traces of evoked EPSCs recorded from autapses under the following conditions: WT, Null, Null + BDNF, Null + BDNF + neutralization. (**C**) Bar graph shows mean evoked EPSC amplitude. (**D**) Representative traces of average current response to 5 s application of 500 mM sucrose. Experimental groups same as (**B**). (**E–G**) Bar graphs show mean RRP size (**E**), $P_{vr}$ (**F**) and 25 ms ISI – PPR (**G**). (**H**) Experimental scheme depicting exogenous application of BDNF in *Mecp2^{Null/y}* neurons, showing neuronal membrane (black), BDNF (red) and TrkB receptor (blue). (**I–M**) Bar graphs show mean evoked EPSC amplitude (**I**), RRP size (**J**), dendrite length (**K**), glutamatergic synapse number (**L**) and neuronal soma area (**M**). Number of neurons (n) shown in the bars for all graphs. Data shown as mean ± SEM. *$p<0.05$; **$p<0.01$; ***$p<0.001$; ns: not significant.

The following figure supplements are available for figure 3:

**Figure supplement 1.** NGF overexpression in *Mecp2^{Null/y}* neurons does not rescue glutamatergic synaptic output.

**Figure supplement 2.** Exogenous application of BDNF does not alter release efficiency or short-term plasticity in *Mecp2^{Null/y}* hippocampal neurons.

## BDNF overexpression restores synapse density in *Mecp2^{Null/y}* glutamatergic neurons

To verify if determining glutamatergic synapse number by estimating presynaptic VGLUT1+ puncta was correlated to identifying both pre- and postsynaptic colocalized puncta, we labeled WT and *Mecp2^{Null/y}* neurons with Homer1, VGLUT1 and MAP2 under different experimental conditions (*Figure 4A*). First, we evaluated the density of VGLUT1 and Homer1 synaptic markers and found that MeCP2-deficient neurons displayed significant reduction in density of both markers, which were rescued upon BDNF overexpression and exogenous BDNF application (*Figure 4B and C*). We also observed that VGLUT1 and Homer1 densities were reverted back to *Mecp2^{Null/y}* levels upon BDNF neutralization (*Figure 4B and C*). Next, we assessed the density of functional synapses by estimating the density of VGLUT1-Homer1 colocalized synaptic puncta. We found 41% decrease in VGLUT1-Homer1 puncta density in *Mecp2^{Null/y}* neurons, which was restored up to 93 and 87% upon BDNF overexpression and exogenous BDNF application, respectively (*Figure 4D*). Additionally, the rate of colocalization of VGLUT1 with Homer1 was significantly reduced upon loss of MeCP2 (20% decrease) and restored up to 99 and 95% specifically via BDNF overexpression and exogenous BDNF, respectively (*Figure 4E*). Finally, we also examined the expression levels of VGLUT1 and Homer1 and found that intensities of both markers remained unchanged across all experimental conditions (*Figure 4—figure supplement 1A–C*).

## BDNF-induced TrkB activation regulates glutamatergic synapse number in a cell autonomous and autocrine manner

We have observed so far that the loss-of-function phenotype seen in *Mecp2^{Null/y}* neurons is reminiscent of that of conditional BDNF knockout mice (*Chang et al., 2006*), and that physiological and morphological deficits are normalized upon BDNF overexpression as well as exogenous application, at single-cell level. If BDNF deficiency is responsible for the synaptic and growth deficits in *Mecp2^{Null/y}* glutamatergic neurons, does paracrine signaling play a role in ameliorating loss of synapse number and function? This is particularly relevant for RTT patients wherein neighboring WT neurons may be impaired in their ability to restore loss of synapse number in *Mecp2^{Null/y}* neurons despite mosaic expression of WT and mutant MeCP2. In order to address this question, we designed an *in vitro* RTT model to examine putative paracrine BDNF activity. We co-cultured WT and *Mecp2^{Null/y}* hippocampal neurons and labeled neurons and corresponding synapses. For this purpose, we pre-incubated neurons of either genotype with distinct lentiviral constructs expressing Synaptophysin-GFP tagged to nucleus-localized RFP (WT) and Synaptophysin-mKate tagged to nucleus-localized GFP (*Mecp2^{Null/y}*) to identify origin of synapses (*Figure 5A*-left panel). Co-labeling for VGLUT1 puncta and MAP2 enabled identifying glutamatergic synapses and their dendritic localization respectively, and synapse density was analyzed by estimating the number of colocalized Syp+ VGLUT1+ puncta. (*Figure 5B*). We found that *Mecp2^{Null/y}* neurons showed 34% decrease in number of

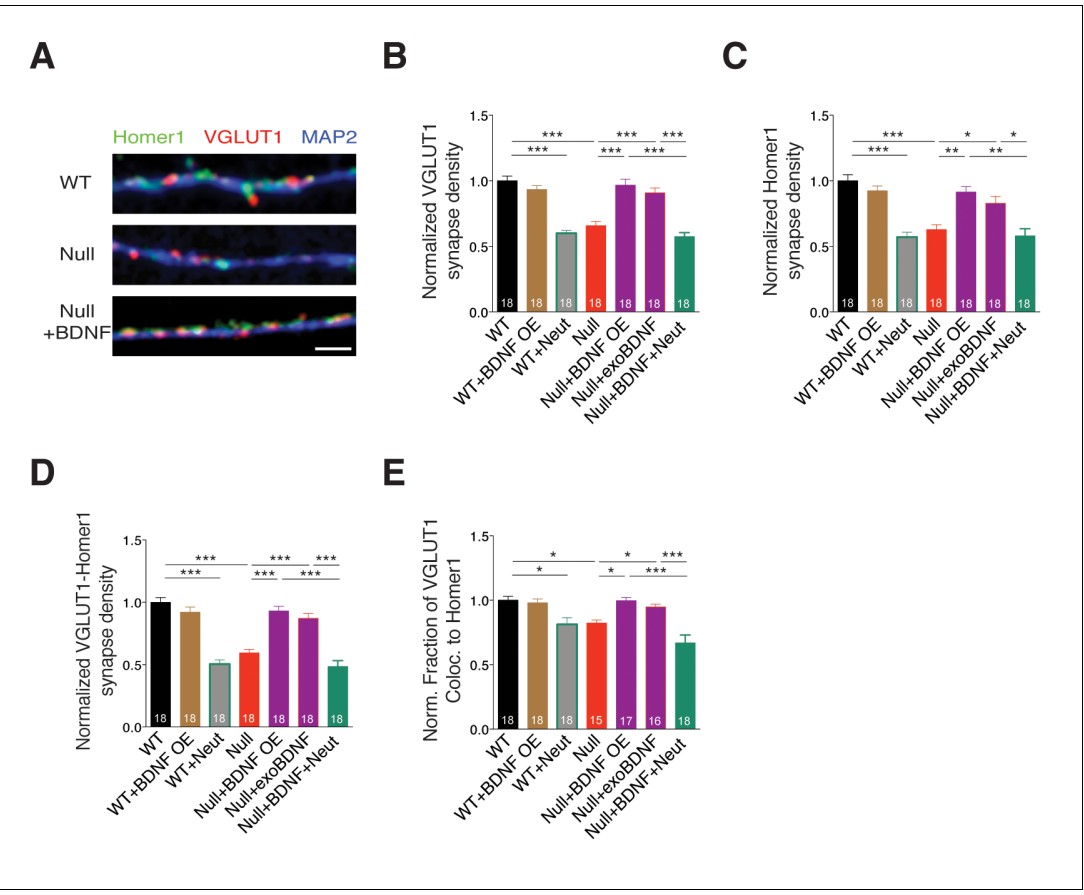

**Figure 4.** BDNF overexpression in *Mecp2*$^{Null/y}$ glutamatergic neurons restores synapse density. (**A**) Representative images of WT and *Mecp2*$^{Null/y}$ neurons labeled with MAP2, VGLUT1 and Homer1 under the following conditions (from top to bottom): WT, Null, Null + BDNF. Scale bar represents 3 μm. (**B–E**) Bar graphs show mean normalized VGLUT1 synapse density (**B**), Homer1 synapse density (**C**), colocalized VGLUT1-Homer1 synapse density (**D**), and normalized fraction of VGLUT1 puncta colocalized to Homer1 (**E**). Number of neurons (n) shown in the bars for all graphs. Data shown as mean ± SEM. *p<0.05; **p<0.01; ***p<0.001.

The following figure supplement is available for figure 4:

**Figure supplement 1.** BDNF overexpression in *Mecp2*$^{Null/y}$ glutamatergic neurons does not alter expression levels of synaptic markers.

glutamatergic synapses (*Figure 5B*-left panel and **C**), which was also consistent with the data from single cells (*Figures 2* and *3*) and *in vivo* (*Chao et al., 2007*).

We next examined the impact of BDNF overexpression in *Mecp2*$^{Null/y}$ neurons to probe their ability to restore synaptic deficit (*Figure 5A*-right panel). To achieve BDNF overexpression, we pre-incubated WT and *Mecp2*$^{Null/y}$ neurons with distinct lentiviral constructs, expressing both BDNF and Synaptophysin-GFP (WT) or Synaptophysin-mKate (*Mecp2*$^{Null/y}$) tagged to nucleus-localized RFP or GFP respectively, via P2A and T2A self-cleaving peptides (Refer Materials and methods). Overexpressing BDNF specifically in WT neurons did not have an impact on number of WT synapses or those formed by proximate *Mecp2*$^{Null/y}$ control neurons with the latter still displaying a 35% decrease in synapse number (*Figure 5B*-middle panel and **C**). Only overexpression of BDNF specifically in *Mecp2*$^{Null/y}$ neurons when co-cultured with WT neurons enhanced number of *Mecp2*$^{Null/y}$ synapses up to 99% of WT levels (*Figure 5B*-right panel and **C**); reaffirming that increased BDNF levels specifically in RTT-like neurons enabled rescue of synaptic phenotype. We then asked if BDNF overexpression also impacted nuclei size in *Mecp2*$^{Null/y}$ neurons. *Mecp2*$^{Null/y}$ neuronal nuclei were 23% and 27% smaller in size when co-cultured with WT and BDNF overexpressing WT glutamatergic

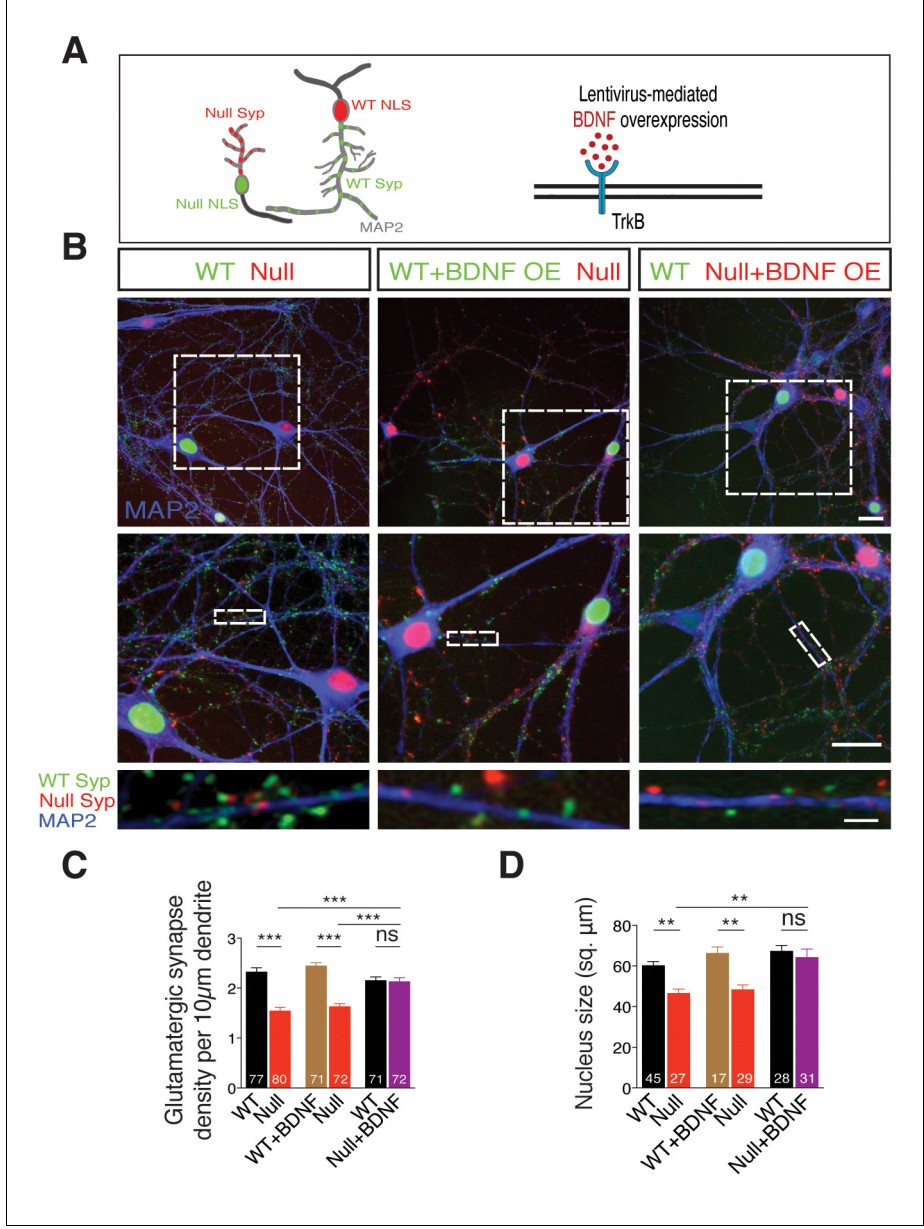

**Figure 5.** TrkB activation specifically in *Mecp2*$^{Null/y}$ neurons in an *in vitro* RTT model normalizes glutamatergic synapse number in a cell autonomous and autocrine manner. (**A**) Left panel: Neuronal pair depicting lentivirus-mediated labeling of nucleus (NLS) and synapses of WT (NLS: red, Synaptophysin: green) and *Mecp2*$^{Null/y}$ neurons (NLS: green, Synaptophysin: red), co-stained for MAP2 to identify synapses localized on dendrites. Right panel: Experimental scheme of lentivirus-mediated BDNF overexpression in WT or *Mecp2*$^{Null/y}$ neurons, showing neuronal membrane (black), BDNF (red) and TrkB receptor (blue). (**B**) Representative images of co-cultured WT and *Mecp2*$^{Null/y}$ neurons labeled with MAP2 under the following conditions (from left to right): WT/Null; WT+BDNF/Null and WT/Null+BDNF. Bottom panel shows WT (green) and Null (red) Synaptophysin+ synapses localized on a single dendrite. Scale bars on top and middle panel represent 20 μm. Scale bar on bottom panel represents 3 μm. (**C,D**) Bar graphs show mean glutamatergic synapse density (**C**) and nucleus size (**D**) for all co-cultured groups. Number of neurons (n) shown in the bars for all graphs. Data shown as mean ± SEM. **p<0.01; ***p<0.001; ****p<0.0001; ns: not significant.

neurons, respectively (*Figure 5D*). However, BDNF overexpression specifically in *Mecp2$^{Null/y}$* neurons restored nucleus size up to 96% of that of co-cultured WT neurons (*Figure 5D*). These findings led us to two major conclusions. (i) Decreased synapse number and nucleus size in *Mecp2$^{Null/y}$* neurons *per se* indicate a cell autonomous role for MeCP2 in regulating normal growth in glutamatergic neurons (*Belichenko et al., 2009*; *Blackman et al., 2012*; *Kishi and Macklis, 2010*). (ii) Restoration of both glutamatergic synapse number and nucleus size upon BDNF overexpression specifically in *Mecp2$^{Null/y}$* neurons is best explained by an autocrine and highly focal effect for BDNF in regulating glutamatergic synapse formation and neuronal size.

## Application of a TrkB agonist restores glutamatergic synapse number in *Mecp2$^{Null/y}$* neurons

Given that restoration of autocrine BDNF secretion in itself is sufficient for normal growth of *Mecp2$^{Null/y}$* neurons and that neighboring WT neurons are unable to support TrkB signaling or normalize synapse number, it is still possible to exogenously activate TrkB receptors on *Mecp2$^{Null/y}$* neurons and restore glutamatergic synapse number. To test this, we repeated the above experiment in the presence of a TrkB agonist 7,8-dihydroxyflavone (7,8-DHF) (*Figure 6A*) that binds to its extracellular domain and activates TrkB-mediated downstream signaling (*Jang et al., 2010*). Intriguingly, application of 500 nM 7,8-DHF at DIV 6, 9 and 12 equalized synapse number in all experimental groups. *Mecp2$^{Null/y}$* synapses were restored back up to 94% and 97% of WT numbers when co-cultured with either WT neurons or WT neurons overexpressing BDNF, respectively (*Figure 6B*). As found in the single-cell system, 7,8-DHF did not have an effect on WT neurons overexpressing BDNF. Similarly, *Mecp2$^{Null/y}$* synapses already overexpressing BDNF remained unaltered indicating that sufficient TrkB stimulation was already prevalent in these neurons (*Figure 6B*). These data proved that 7,8-DHF was able to bypass BDNF synthesis deficit and exogenously trigger TrkB activation in *Mecp2$^{Null/y}$* neurons and augment synapse formation in a cell autonomous manner.

These findings reveal a crucial mechanism wherein BDNF has an autocrine and cell autonomous role, potentially activating TrkB receptors only on the same neuron via a positive feed forward system, thereby regulating excitatory synapse formation and normal growth of *Mecp2$^{Null/y}$* neurons.

## BDNF controls glutamatergic synapse number acting as a presynaptic rate-limiting factor in a MeCP2-dependent manner

We then asked if loss of BDNF activity affects glutamatergic synapse formation through impairment of pre- or postsynaptic function. To address this question, we analyzed WT and *Mecp2$^{Null/y}$* glutamatergic synapse densities, as described before, from proximal dendrites of identified WT and *Mecp2$^{Null/y}$* postsynaptic neurons (*Figure 6C*). We reasoned that in case of a presynaptic mechanism, *Mecp2$^{Null/y}$* glutamatergic synapse densities would remain unchanged whether formed onto a WT or *Mecp2$^{Null/y}$* postsynaptic neuron. Indeed, we found that the densities of *Mecp2$^{Null/y}$* glutamatergic synapses were similarly (56 and 45%) decreased when formed onto WT as well as *Mecp2$^{Null/y}$* postsynaptic neurons, respectively (*Figure 6D*-left). By contrast, in case of presynaptic *Mecp2$^{Null/y}$* neurons overexpressing BDNF, densities of Null glutamatergic synapses made onto WT and *Mecp2$^{Null/y}$* postsynaptic neurons were restored up to 87 and 86% of WT levels (*Figure 6D*-right). This analysis strongly suggests that BDNF deficiency in cultured glutamatergic neurons reduces overall synaptic output without impairing the ability to receive glutamatergic synaptic input.

## Neurons lacking MeCP2 have smaller somata and nuclei as compared to WT in hippocampal CA1 of *Mecp2$^{+/-}$* mice

The autocrine role of BDNF secretion in an *in vitro* RTT model might argue that in heterozygous mice with mosaic MeCP2 expression patterns mimicking RTT, impaired BDNF synthesis may contribute to a persistent growth deficit specifically in neurons lacking MeCP2 *in vivo*. In addition to smaller somata in *Mecp2$^{Null/y}$* neurons, reduced neuronal nuclei size has been reported across previous MeCP2 loss-of-function studies (*Chen et al., 2001*; *Rietveld et al., 2015*) that has been shown to be rescued by administering 7,8-DHF to *Mecp2$^{Null/y}$* mice (*Johnson et al., 2012*). Hence, we looked at the soma area and nucleus size of CA1 neurons in two- and eight-week old *Mecp2$^{+/-}$* mice to examine if there is a persistent phenotype in MeCP2-deficient neurons. For this purpose, we labeled the hippocampal CA1 for MeCP2, MAP2 and DAPI to perform quantitative analysis of MeCP2-positive

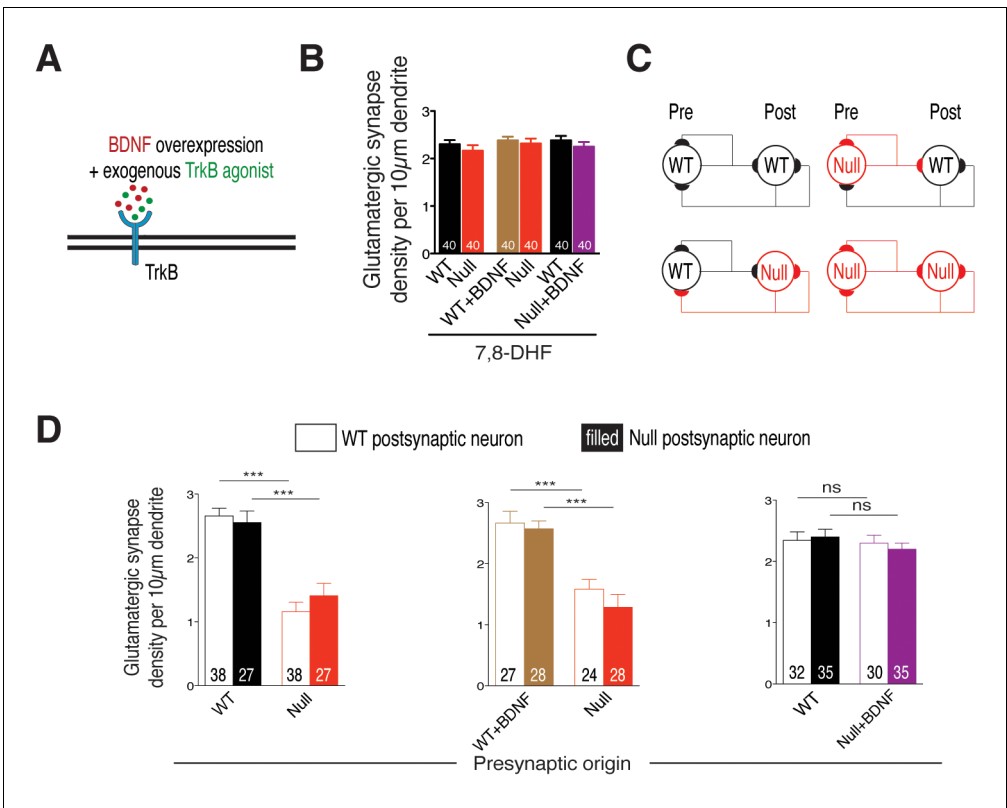

**Figure 6.** Cell autonomous BDNF-TrkB signaling regulates glutamatergic synapse number by functioning as a presynaptic rate-limiting factor. (**A**) Experimental scheme of lentivirus-mediated BDNF overexpression and exogenous application of TrkB agonist in WT or *Mecp2*[Null/y] neurons, showing neuronal membrane (black), BDNF (red), TrkB receptor (blue) and TrkB agonist (green). (**B**) Bar graph shows glutamatergic synapse density for all co-cultured groups upon application of TrkB agonist, 7,8-DHF. (**C**) Example two-neuron scheme illustrating pre- and postsynaptic neurons (WT or *Mecp2*[Null/y]) forming synapses onto itself or onto the partner neuron, in a mixed WT/ Null co-culture system. (**D**) Bar graphs show glutamatergic synapse density measured from a postsynaptic WT (clear bars) and Null (filled bars) neuron for all experimental groups. Number of neurons (n) shown in the bars for all graphs. Data shown as mean ± SEM. ***$p<0.001$; ns: not significant.
The following figure supplement is available for figure 6:

**Figure supplement 1.** WT and *Mecp2*[Null/y] hippocampal neurons reveal transient and persistent fusion events upon stimulation, show equivalent activity-dependent BDNF secretion and membrane-resident BDNF-SpH fraction.

and –negative neuronal nuclei (*Figure 7A and B*). Strikingly, we found that MeCP2-negative neuronal somata were 20 and 14% smaller than that of MeCP2-positive neurons in two- (*Figure 7C*) and eight-week old (*Figure 7D*) *Mecp2*[+/-] mice respectively. Similarly, MeCP2-negative neuronal nuclei were 18 and 21% smaller than MeCP2-positive neurons in two- (*Figure 7E*) and eight-week old (*Figure 7F*) *Mecp2*[+/-] mice respectively. These *in vitro* and *in vivo* findings together validate a cell autonomous effect and autocrine role for BDNF secretion in *Mecp2*[Null/y] neurons and lead to promising avenue for future investigations studying cell autonomous effects of BDNF and MeCP2.

## Discussion

Previous MeCP2 loss-of-function studies have shown decrease in glutamatergic synapse number and synapse density as well as reduced dendritic complexity and arborization associated with loss of MeCP2 (*Belichenko et al., 2009*; *Chao et al., 2007*; *Jentarra et al., 2010*; *Larimore et al., 2009*; *Zhou et al., 2006*) but underlying mechanisms are not fully understood. In our study, we find a

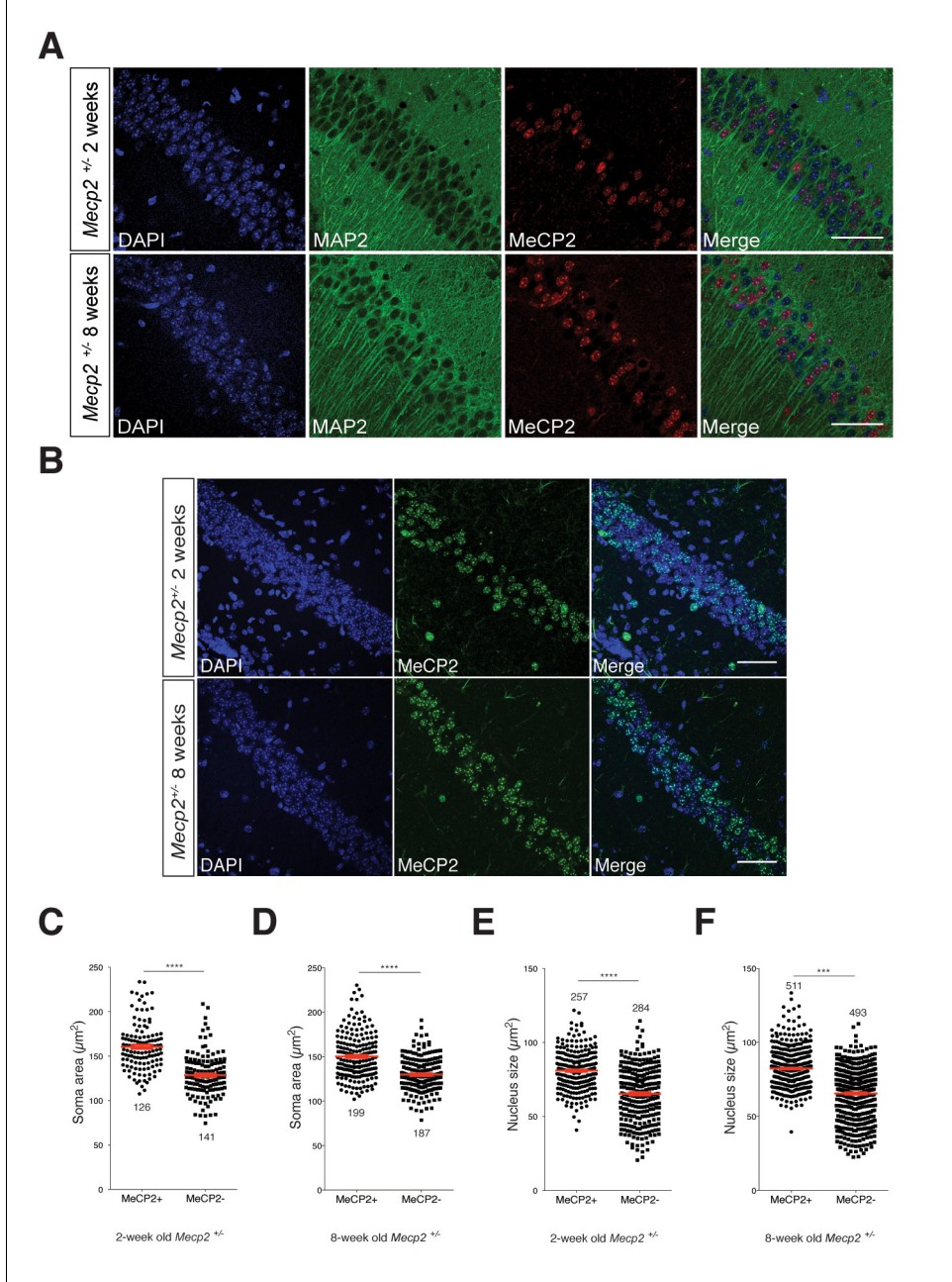

**Figure 7.** MeCP2-deficient hippocampal CA1 neurons have smaller somata and nuclei in *Mecp2$^{+/-}$* heterozygous mice *in vivo*. (A) Representative images of hippocampal CA1 from *Mecp2$^{+/-}$* mice labeled for DAPI (blue), MAP2 (green) and MeCP2 (red) at two- (top) and eight-weeks of age (bottom). Scale bar represents 50 µm. (B) Representative images of hippocampal CA1 from *Mecp2$^{+/-}$* mice labeled for DAPI (blue) and MeCP2 (green) at two- (top) and eight-weeks of age (bottom). Scale bar represents 50 µm. (C–F) Scatter plots show mean soma area and nucleus size of MeCP2+ and MeCP2- neurons at two- (C,E) and eight-weeks of age (D,F). Number of neurons (n) shown in all graphs. Data shown as mean ± SEM. ***p<0.001; ****p<0.0001.

generalized growth and differentiation defect in glutamatergic *Mecp2$^{Null/y}$* neurons with MeCP2-deficient neurons displaying reduced neurite outgrowth and soma area (*Figure 1*). In examining causal mechanisms, we find a critical role for BDNF. The specific loss in neuronal growth can be attributed to suppression of BDNF synthesis since both synaptic output and morphological pheno-types could be restored by lentivirus-mediated and exogenous application of BDNF, in a TrkB-

dependent manner (*Figure 2*). Moreover, we show that blocking BDNF-TrkB signaling in wildtype neurons leads to a MeCP2-deficient state, strongly arguing for nearly complete suppression of the canonical BDNF-TrkB pathway in mutant neurons. Further, we demonstrate that the BDNF-mediated growth pathway is strictly cell autonomous since (i) neighboring wildtype neurons fail to rescue *Mecp2*$^{Null/y}$ neuronal morphological deficits in a RTT model *in vitro* (*Figure 5*), and (ii) MeCP2-deficient neurons have smaller somata as well as nuclei in both two- and eight-week old *Mecp2*$^{+/-}$ mice *in vivo* (*Figure 7*), given the reduced BDNF protein levels in *Mecp2*$^{+/-}$ mice (*Wang et al., 2006*) and reduced mRNA levels in postmortem brain samples from RTT patients (*Abuhatzira et al., 2007*; *Deng et al., 2007*). This cell autonomy further illustrates that a change in circuit activity is unlikely to explain synapse loss associated with MeCP2-negative neurons in RTT models.

Much attention has been received towards manipulating either MeCP2 levels or protein levels of MeCP2-regulated genes predominantly because reactivation of the *Mecp2* or target gene has been shown to rescue morphological phenotypes, behavior and overall neurological function (*Chang et al., 2006*; *Chao et al., 2007*; *Guy et al., 2007*; *Kline et al., 2010*; *Larimore et al., 2009*; *McGraw et al., 2011*; *Nguyen et al., 2012*). The fact that several synaptic defects in *Mecp2*$^{Null/y}$ neurons might be fundamentally linked to simply BDNF synthesis despite MeCP2 regulating transcription of a myriad of target genes renders MeCP2-BDNF interaction extremely pivotal. Along these lines, endogenous MeCP2 knockdown reduced dendritic length in E18 hippocampal neurons, which was fully rescued upon BDNF overexpression (*Larimore et al., 2009*), which is in agreement with our rescue data (*Figure 2C*). Differential BDNF function based on brain region and neuronal subtypes has been emphasized across various studies. In particular, reduced BDNF levels in the brainstem of *Mecp2*$^{Null}$ mice increased the amplitude of spontaneous and evoked EPSCs in nTS (nucleus tractus solitarius) neurons, which were fully rescued upon exogenous BDNF application (*Kline et al., 2010*). This is comparable to the BDNF-mediated rescue of synaptic function in glutamatergic *Mecp2*$^{Null/y}$ neurons in our study (*Figure 2F–I*).

That BDNF is a neurotrophin with poor blood-brain barrier penetration characteristics (*Poduslo and Curran, 1996*) prompted the need for small molecule substitutes that act as TrkB agonists and activate TrkB signaling (*Jang et al., 2010*; *Massa et al., 2010*). Administration of TrkB agonists (*Johnson et al., 2012*; *Kron et al., 2014*; *Schmid et al., 2012*) and other candidates including CX546 (*Ogier et al., 2007*), environmental enrichment (*Kondo et al., 2008*; *Lonetti et al., 2010*), insulin-like growth factor-1 (*Tropea et al., 2009*), cysteamine (*Roux et al., 2012*) and fingolimod (*Deogracias et al., 2012*) to MeCP2 mutant mice ameliorated several behavioral and functional RTT phenotypes both *in vitro* and *in vivo*. In our unique WT/Null RTT-like model, 7,8-DHF application clearly substantiated the autocrine and cell autonomous function of BDNF as well as BDNF-TrkB feed-forward loop impairment in *Mecp2*$^{Null/y}$ neurons (*Figures 5C* and *6B*). Impaired BDNF-TrkB activity could be due to differential surface expression of postsynaptic TrkB receptors and therefore inadequate availability of surface TrkB for binding to BDNF. However, complete rescue of both physiological and morphological phenotypes upon BDNF overexpression (*Figure 2*) eliminated this possibility and highlighted a crucial upstream deficit in BDNF synthesis and release. Further studies are required to elucidate how 7,8-DHF functions to act *in vitro* and *in vivo* to ameliorate RTT symptoms and enable normalization of glutamatergic synapse numbers.

BDNF synthesis deficit in MeCP2 lacking neurons has resulted in different phenotypic effects across various studies. In particular, Wang and colleagues reported decrease in absolute amounts of BDNF released but also an increase in percentage of BDNF content available for spontaneous release in *Mecp2*$^{Null/y}$ nodose ganglion (NG) neurons (*Wang et al., 2006*). This may argue for cell-type specific effects of BDNF. Alternatively in our study, we demonstrate that glutamatergic synaptic output is normalized in *Mecp2*$^{Null/y}$ hippocampal neurons upon restoring BDNF levels (*Figures 2* and *3*) and that BDNF neutralization experiments indicate a defect in synthesis (*Figure 3*), which suggest decreased availability of BDNF for release. By contrast, BDNF could also be functioning through other parallel pathways. For example, a study based on hippocampal slices demonstrated impaired activity-dependent endogenous BDNF release from presynaptic mossy fibers onto CA3 pyramidal neurons of symptomatic MeCP2 mutant mice that was correlated to impaired TRPC3 signaling (*Li et al., 2012*).

BDNF and other neurotrophins bind to the cell surface due to their highly positive charge at physiological pH (*Blöchl and Thoenen, 1996*; *Brigadski et al., 2005*; *de Wit et al., 2009*), and locally modulate developing synapses by spatially restricting BDNF action (*Zhang and Poo, 2002*).

Additionally, several peptidergic hormones and secretory molecules including Wnt (*Cadigan et al., 1998*; *Papkoff and Schryver, 1990*), Semaphorin 3A (*Bouzioukh et al., 2006*; *De Wit et al., 2005*; *Sahay et al., 2005*) and BDNF (*de Wit et al., 2009*) have been shown to remain membrane resident after secretion. Indeed, we verified this hypothesis by examining the fate of BDNF secretion in WT and *Mecp2$^{Null/y}$* hippocampal neurons using the BDNF-supereclliptic pHluorin (BDNF-SpH) live-cell imaging assay (*Kolarow et al., 2007*) (*Figure 6—figure supplement 1*) (Refer Materials and methods). We measured activity-dependent BDNF release by 60 mM KCl-induced depolarization, and quantified surface fraction of BDNF-SpH by application of pH 5.5 MES-buffered acid solution. We found that a significant fraction of BDNF-SpH remained membrane-resident as evident from MES-buffered acid quenching experiments (*Figure 6—figure supplement 1B and G*), without affecting activity-induced BDNF secretion in both WT and *Mecp2$^{Null/y}$* neurons (*Figure 6—figure supplement 1D–F*). This complements our previous findings and shows that stable membrane deposits of BDNF could potentiate TrkB receptor activation cell autonomously and drive glutamatergic synapse number by confining activity to its own neurites. Additionally, the fact that activity-dependent regulated BDNF release was unaffected in MeCP2-deficient glutamatergic neurons suggested that the observed synaptic phenotypes were accounted for by a deficit in constitutive BDNF release. Further studies specifically measuring pre- and postsynaptic release of BDNF from WT and *Mecp2$^{Null/y}$* cultured glutamatergic neurons are essential to determine the locus of activity-dependent regulation of excitatory synapses.

What is the functional purpose of this autocrine feed-forward signaling loop? There seems to be a link between intrinsic activity and growth in WT neurons by allowing BDNF to work on its own neurons, coupling neuronal firing patterns to their basic growth properties. This further implies that reduced activity in *Mecp2$^{Null/y}$* neurons (*Chang et al., 2006*; *Chao et al., 2007*; *Dani et al., 2005*; *Tropea et al., 2009*) may disrupt activity-dependent neuronal growth. This deficiency could be repaired by supplying an additional source of BDNF (or TrkB agonist) that activates WT-like TrkB signaling. BDNF has been suggested to act as an autocrine factor mainly in regulating dendrite development in adult-born granule cells (GCs) (*Wang et al., 2015*) and promoting axon formation in embryonic hippocampal neurons (*Cheng et al., 2011*). In RTT pathophysiology in particular, we show for the first time that this localized autocrine function and cell autonomy of BDNF-TrkB together offer a plausible mechanism and explain why MeCP2 mosaicism in RTT females is ineffective in restoring phenotypes at a cellular, network or behavioral level. It may be possible that BDNF released could be less efficient in reaching neighboring neurons in co-cultures; however, the cell autonomous effect observed in autaptic neurons *in vitro*, and *in vivo*, as well as the autocrine BDNF dependence seen *in vitro*, argue against putative BDNF paracrine effects. However, what remains to be understood is if the predominant impact of impaired BDNF signaling on circuitry stems from the pre- or postsynaptic site. Initial morphological analysis showed axonal as well as dendritic growth deficiency (*Figure 1*). Further analysis in mixed co-culture experiments demonstrated a presynaptic deficit wherein densities of *Mecp2$^{Null/y}$* glutamatergic synapses onto the postsynaptic neuron remained independent of genotype (*Figure 6D*). This points towards a major 'propagation' defect specifically in *Mecp2$^{Null/y}$* neurons in a mosaic circuit wherein they receive synaptic input similar to WT neurons. However, MeCP2-deficient glutamatergic neurons show reduced capacity to communicate output signal to a postsynaptic neuron thereby reducing efficiency of innervation across specific brain regions and dampening synaptic output. In fact, one could postulate that glutamatergic neurons lacking MeCP2 are essentially not 'deaf' but 'mute'. Interestingly, there is evidence of non-cell autonomous mechanisms contributing to neuronal development (*Braunschweig et al., 2004*) and dendritic arborization of cortical pyramidal neurons (*Kishi and Macklis, 2010*). In a mosaic brain, MeCP2-deficient mute neurons may contribute to reduced overall activity, which in turn could reduce activity-dependent BDNF secretion and thereby result in putative non-cell autonomous effects as well. Hence, future studies comparing wildtype and heterozygous *Mecp2$^{+/-}$* mice may help reveal additional non-cell autonomous effects *in vivo*.

It will be important to determine how the proposed presynaptic BDNF-mediated mechanistic pathway relates to other cell types, as well as to investigate specific downstream signaling pathways that might be affected as a result of BDNF synthesis deficiency. For example, MeCP2-deficient GABAergic neurons show decrease in inhibitory quantal size with reduced Gad1 and Gad2 levels (*Chao et al., 2010*) while BDNF released from postsynaptic target neurons acts locally to modulate GABAergic synapse formation (*Kohara et al., 2007*). More generally, besides providing a

mechanistic role for BDNF in MeCP2 mutant neurons, our results substantiate the significance of manipulating BDNF-TrkB interaction as a potential therapeutic strategy in alleviating the course of the RTT syndrome.

## Materials and methods

### Animals and cell culture

All procedures to maintain and use mice were approved by the Animal Welfare Committee of Charité Medical University and the Berlin State Government. $Mecp2^{Null/y}$ male mice on a C57BLJ6/N background and/or $Mecp2^{Tg1}$ male mice on a FVB/N background were used for morphological and electrophysiological studies.

Primary hippocampal neurons were prepared from P0-P2 newborn mice and cultured on astrocyte feeder layers. First, WT astrocytes derived from P0-P2 mouse cortices were plated on collagen/poly-D-lysine microislands made on agarose-coated coverslips using a custom-made rubber stamp. For autaptic neuron studies, $Mecp2^{Null/y}$ or $Mecp2^{Tg1}$ neurons were cultured along with their littermate controls, plated at a density of 3000 neurons per 35 mm well and grown in Neurobasal-A media containing B-27 supplement and Glutamax (Invitrogen, Germany). For experiments utilizing continental cultures, glial proliferation was arrested by adding the antimitotic agent 8 μM 5-fluoro-2-deoxyuridine and 20 μM uridine (FUDR) to the astrocyte media. For experiments involving co-culture of WT and $Mecp2^{Null/y}$ neurons, cells of either genotype were incubated with lentiviral constructs of interest for 1.5 hr at 37°C on a rotating wheel. Cells were then centrifuged at 1500 rpm for 5 min twice to remove any viral debris after which WT and $Mecp2^{Null/y}$ neurons were plated at a density of 14,000 neurons each per 22 mm well. This procedure enables identification of synapses formed by either genotype under desired conditions. For all experiments, neurons were incubated at 37°C for 12–14 d before subjecting them to further analysis.

### Lentiviral constructs

Mouse cDNA constructs of pre-proBDNF, NGF and BDNF-SpH rat cDNA (kindly provided by Prof. Matthijs Verhage, Center for Neurogenomics and Cognitive Research, Amsterdam, The Netherlands) were cloned into vectors under control of the neuron-specific synapsin promoter. For co-culture experiments, we used a synapsin promoter-driven lentiviral shuttle vector expressing BDNF, cloned N-terminally to a self-cleaving T2A peptide of an expression cassette, which harbors (i) a nuclear localization sequence-tagged green fluorescent protein (NLS-GFP) or red fluorescent protein (NLS-RFP), which was fused C-terminally via a self-cleaving P2A peptide (*Kim et al., 2011*) to (ii) Synaptophysin-mKate or Synaptophysin-GFP respectively (NLS-GFP-P2A-SypmKate-T2A-BDNF or NLS-RFP-P2A-SypGFP-T2A-BDNF). Control lentiviral vectors included NLS-GFP-P2A-SypmKate and NLS-RFP-P2A-SypGFP.

Lentiviral vectors and production were based on previously published protocols (*Lois et al., 2002*). The production was done by the Viral-Core-Facility of the Charité – Universitaetsmedizin Berlin. Briefly, HEK293T cells were cotransfected with 10 μg of shuttle vector and helper plasmids (pCMVdR8.9 and pVSV-G - 5 μg each) with X-tremeGENE 9 DNA transfection reagent (Roche Diagnostic, Switzerland). Virus-containing cell culture supernatant was harvested 72 hr post transfection and purified by filtration to remove cellular debris. Filtrate aliquots were flash-frozen in liquid nitrogen and stored at −80°C. Viral titer for all rescue experiments was determined using WT hippocampal continental neuronal cultures. Autaptic and continental hippocampal neurons were infected with lentivirus expressing BDNF or NGF on DIV 2. In case of co-culture experiments, WT and $Mecp2^{Null/y}$ neurons were incubated with lentivirus expressing tagged-BDNF and plated as described above.

### Drug treatment

Recombinant human BDNF (Promega, Madison, WI) was exogenously applied to the autaptic neurons at DIV 2 and added every 2–3 days at 50 ng/ml. The TrkB receptor antagonist ANA-12 (Tocris, Germany) was applied to neurons through culture medium at DIV 6 and added every three days at 10 μM. The BDNF neutralizing antibody α-BDNF (Millipore, Germany) was applied to neurons through culture medium at DIV 6 and added every three days at a dilution of 1:100. Neurons were

treated with the TrkB agonist 7,8-DHF (Sigma Aldrich, Saint Louis, MO) at DIV 6, 9 and 12 at a working concentration of 500 nM.

## Electrophysiology

Whole-cell voltage-clamp recordings were obtained from autaptic neurons at DIV 12–17. Currents were recorded from neurons held at −70 mV with a Multiclamp 700B amplifier (Molecular Devices, Sunnyvale, CA) under the control of Clampex 9.2 (Molecular Devices). Data were sampled at 10 kHz and low-pass Bessel filtered at 3 kHz. Series resistance was compensated at 70% and only cells with <12 MΩ resistance were included. In general, an approximately equal number of cells were recorded from all groups on a given experimental day and data from at least two independent cultures were analyzed per experiment.

Neurons were placed in standard extracellular solution, 300 mOsm pH 7.4, containing 140 mM NaCl, 2.4 mM KCl, 10 mM HEPES, 10 mM glucose, 2 mM $CaCl_2$ and 4 mM $MgCl_2$. The patch pipette internal solution, 300 mOsm pH 7.4, contained 136 mM KCl, 17.8 mM HEPES, 1 mM EGTA, 0.6 mM $MgCl_2$, 4 mM ATP-Mg, 0.3 mM GTP-Na, 12 mM phosphocreatine, and 50 U/ml phosphocreatine kinase. Hypertonic sucrose solution was prepared as 500 mM sucrose in standard extracellular solution (*Rosenmund and Stevens, 1996*).

Excitatory postsynaptic currents (EPSCs) were evoked by briefly depolarizing neurons from −70 mV to 0 mV for 2 ms. Application of hypertonic sucrose solution for 5 s was facilitated using a fast-flow system triggering release of the readily releasable pool (RRP) characterized by a transient inward current. RRP charge was estimated by integrating the total area under the transient curve obtained. Vesicular release probability ($P_{vr}$) was determined by calculating ratio of EPSC charge over RRP charge. Short-term plasticity was determined by evoking 2 EPSCs with an inter-stimulus interval of 25 ms to measure paired-pulse ratio (PPR). PPR was determined by calculating the ratio of EPSC amplitude of second over the first synaptic response.

Electrophysiological data were analyzed offline using Axograph X (Axograph Scientific, Berkeley, CA), Excel (Microsoft, Redmond, WA) and Prism (GraphPad, La Jolla, CA). Unless specified otherwise, statistical significance was determined using one-way ANOVA with Tukey *post hoc* test (for three or more groups with normal distribution) or Student's *t test* (for two groups with normal distribution). In case of data not normally distributed according to D'Agostino-Pearson test, statistical significance was determined using non-parametric Kruskal-Wallis test with Dunn's *post hoc* test (for three or more groups) or Mann-Whitney *U* test (for two groups).

## Immunocytochemistry

At DIV 13–15, cells were fixed in 4% paraformaldehyde for 15 min after which they were washed thrice in 1x PBS. After permeabilizing and blocking with 5% normal donkey serum (NDS) in 0.1% PBS-Tween (PBST) for 1 hr, cells were subsequently incubated with antibodies of interest overnight at 4°C. After washing coverslips thrice with 0.1% PBST for 15 min each, primary antibodies were labeled with secondary Alexa-Fluor 405, 488, 555 or 647 (1:500; Jackson, West Groove, PA) antibodies for 1.5 hr at room temperature. Coverslips were then washed twice with 0.1% PBST and twice with 1x PBS for 15 min each after which they were mounted on glass slides with either Mowiol or ProLong Gold Antifade Reagent (Invitrogen). Labeling was done with (i) guinea pig anti-VGLUT1 (1:4000; Synaptic Systems, Germany), (ii) rabbit anti-VGLUT1 (1:4000; Synaptic Systems), (iii) chicken anti-microtubule-associated protein 2 (MAP2) (1:2000; Millipore), (iv) mouse anti-Tau1 (1:1000; Millipore) and (v) guinea pig anti-Homer1 (1:500; Synaptic Systems).

## Quantification of neuronal morphology

First, 16-bit images were acquired on an Olympus IX81 inverted fluorescence microscope at 20x optical magnification with a CCD camera (Princeton MicroMax; Roper Scientific, Trenton, NJ). All images were analyzed using ImageJ software with relevant custom plugins. At least two independent cultures were imaged and analyzed for every experiment. All images were subject to uniform background subtraction and optimal threshold adjustment.

Quantification of VGLUT1+ puncta in autaptic WT or $Mecp2^{Null/y}$ neurons was used as a measure of presynaptic differentiation and estimating glutamatergic synapse number. This was done by identifying and counting all VGLUT1+ fluorescent spots localized on dendritic branches of every neuron

using a custom macro in ImageJ. Quantification of all MAP2-positive processes with NeuronJ plugin was used to determine total dendrite length while measuring cross-sectional area across the MAP2+ cell body enabled estimation of area of neuronal somata. Nucleus cross-sectional area was measured by tracing the outline of the nucleus using NLS+ identified neurons. Total axonal length analysis was done by quantification of all Tau1+ MAP2- processes per neuron using NeuronJ plugin since an overlap of Tau1 labeling was observed in proximal MAP2-positive dendritic branches in several neurons. Pre- and postsynaptic puncta were estimated by manually counting individual VGLUT1+ and Homer1+ as well as double positive synapses on selected dendritic regions of interest. Glutamatergic synapse density in co-cultures was analyzed by manually counting VGLUT1 and synaptophysin double positive synapses on selected dendritic regions of interest.

Unless specified otherwise, statistical significance was determined using one-way ANOVA with Tukey *post hoc* test (for three or more groups with normal distribution) or Student's *t test* (for two groups with normal distribution). In case of data not normally distributed according to D'Agostino-Pearson test, statistical significance was determined using non-parametric Kruskal-Wallis test with Dunn's *post hoc* test (for three or more groups) or Mann-Whitney *U* test (for two groups).

## Immunohistochemistry

Two animals per time point were fixed by transcardial perfusion with phosphate buffered-4% paraformaldehyde. Sectioning was done on a Leica cryostat CM3050 S and 25 μm coronal sections were obtained for both two- and eight-week old *Mecp2*$^{+/-}$ heterozygous mice. Sections were permeabilized with 1% Triton X-100 for 30 min and blocked with 5% NDS, 2% glycine, 0.5% Triton X-100 in PBS for 30 min at room temperature after which they were incubated in mouse anti-MeCP2 (1:500; Sigma) and chicken anti-MAP2 (1:2000; Millipore) overnight at 4°C. The sections were washed thrice with 1x PBS for 15 min each and stained with secondary Alexa 488 (1:500; Jackson) for 1 hr at RT. Sections were again washed thrice with 1x PBS for 15 min each and mounted on glass slides in Pro-Long Gold Antifade reagent with DAPI (Invitrogen).

## Quantification of neuronal somata and nuclei in hippocampal CA1

Images were acquired on a Leica SP8 laser-scanning confocal microscope. Images were acquired as 8-bit images with 63x oil objective at 1024 × 1024 pixel resolution. Three to four sections were imaged per mouse per time point, images were acquired as z-stack with 15–20 optical sections and maximum intensity projections were created using Fiji. Hippocampal CA1 MeCP2+ and MeCP2- neurons were identified based on their nuclear punctate staining and outlines of corresponding neuronal DAPI+ nuclei were manually traced and cross-sectional area measured using Fiji. Neuronal somata were measured by manually tracing MAP2+ cell bodies across different optical sections and cross-sectional area was measured using Fiji.

## BDNF-supereclliptic pHluorin live cell imaging

We utilized an assay to monitor BDNF exocytosis in hippocampal neurons using a construct expressing BDNF tagged with supereclliptic pHluorin (BDNF-SpH) (*de Wit et al., 2009*). BDNF-SpH was expressed in primary hippocampal neurons and characterized using the pHluorin live-cell imaging assay. Neurons were placed in standard extracellular solution as described in case of electrophysiological measurements and 256 × 256 pixel images were acquired on an Olympus IX71 inverted microscope equipped with an Andor iXon back-illuminated CCD camera and Polychrome V Illumination Unit (Till Photonics, Germany) at 60x magnification (numerical aperture 1.2) and a sampling rate of 1 Hz.

Neurons were stimulated by application of 60 mM KCl (prepared in standard extracellular solution) for 30 s using the fast-flow system as mentioned above. 60 mM KCl-induced membrane depolarization of BDNF-SpH-expressing neurons resulted in an increase in fluorescence (ΔF) that resembled a punctate-pattern distribution representative of BDNF exocytosis. Some fusion events decayed abruptly while many others decayed slowly or remained persistent. 50 mM NH$_4$Cl solution was applied for 30 s to neutralize intracellular pH, causing an abrupt increase in fluorescence intensity enabling visualization of all BDNF-SpH containing compartments. Low pH (pH 5.5) solution of 2-(*N*-morpholino) ethanesulfonic acid was applied for 30 s for acid wash experiments, causing acute dimming of all surface resident BDNF-SpH proteins.

## BDNF-supereclliptic pHluorin image analysis

BDNF release events were analyzed from stacks acquired from time-lapse recordings of BDNF-SpH containing vesicle clusters upon application of KCl, NH$_4$Cl, pH 5.5 solutions or standard extracellular solution. BDNF release was analyzed from both synaptic and extrasynaptic sites. Background subtraction was done using Rolling Ball (50 pixel radius, ImageJ) for all image stacks and ROIs were marked to measure $\Delta$F, using ImageJ. $4 \times 4$ pixel regions of interest (ROIs) were identified and peak change in fluorescence ($\Delta$F) in response to KCl-induced stimulation normalized to initial fluorescence (F$_0$ = average of five frames immediately before onset of stimulus) was measured as a function of time and averaged in Axograph X (Axograph). Baseline subtraction was done for each ROI per cell in Axograph X, and $\Delta$F and $\Delta$F/F$_0$ values for all groups were analyzed in Prism (GraphPad). Release events were characterized by their abrupt increase in fluorescence upon application of KCl followed by a rapid or gradual decrease of fluorescence back to baseline levels, and normalized to their corresponding peak NH$_4$Cl events. The same ROIs were probed in recordings upon application of NH$_4$Cl or MES-buffered acid solutions. Acid quenched BDNF fraction was normalized to corresponding peak NH$_4$Cl response, reflecting the non-internalized pool of BDNF vesicles that are either found on the neuronal surface or easily accessible from the extracellular space around the cell membrane.

## Statistics

All statistical analyses were done using Excel (Microsoft) and Prism (Graphpad). Sample size estimation was done as described previously (*Chao et al., 2007*, *2010*). Detailed statistical data are reported in *Supplementary file 1.*

## Acknowledgements

We thank Annegret Felies, Bettina Brokowski, Berit Soehl-Kielczynski, Carola Schweynoch, Katja Poetschke, Rike Dannenberg and Sabine Lenz for technical assistance; and members of the Rosenmund laboratory for discussions. We also thank Prof. Matthijs Verhage for kindly providing the BDNF-SpH cDNA and Prof. Volkmar Lessmann for careful reading of the manuscript.

## Additional information

### Competing interests

CR: Reviewing editor, *eLife*. The other authors declare that no competing interests exist.

### Funding

| Funder | Grant reference number | Author |
|---|---|---|
| Deutsche Forschungsgemeinschaft | SFB 665 | Charanya Sampathkumar<br>Mayur Vadhvani<br>Christian Rosenmund |
| Deutsche Forschungsgemeinschaft | Exc257 | Yuan-Ju Wu<br>Mayur Vadhvani<br>Britta Eickholt<br>Christian Rosenmund |
| Berlin Institute of Health | CRG Congenital Diseases | Charanya Sampathkumar<br>Christian Rosenmund |

The funders had no role in study design, data collection and interpretation, or the decision to submit the work for publication.

### Author contributions

CS, Conception and design, Acquisition of data, Analysis and interpretation of data, Drafting or revising the article; Y-JW, Conception and design, Acquisition of data, Analysis and interpretation of data; MV, Acquisition of data, Analysis and interpretation of data; TT, contributed lentiviral constructs, Contributed unpublished essential data or reagents; BE, Analysis and interpretation of data, Drafting or revising the article; CR, Conception and design, Analysis and interpretation of data, Drafting or revising the article

**Author ORCIDs**

Christian Rosenmund, http://orcid.org/0000-0002-3905-2444

**Ethics**

Animal experimentation: All procedures to maintain and use mice were approved by the Animal Welfare Committee of Charité Medical University and the Berlin State Government (License no. 0220/09).

## Additional files

**Supplementary files**

• Supplementary file 1. Detailed statistical data.

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
