## [Decision Letter]

Thank you for submitting your article "Loss of MeCP2 disrupts cell autonomous and autocrine BDNF signaling in glutamatergic neurons" for consideration by *eLife*. Your article has been reviewed by three peer reviewers, one of whom, Sacha B Nelson (Reviewer #1), is a member of our Board of Reviewing Editors and the evaluation has been overseen by a Senior Editor.

The reviewers have discussed the reviews with one another and the Reviewing Editor has drafted this decision to help you prepare a revised submission.

It is *eLife* policy that titles should convey some details of the system under study. In this case it might be helpful to add the word "mouse" or "mammalian" to the title.

Summary:

Sampathkumar, Rosenmund, and colleagues use primary neuronal cultures, genetics, pharmacology and electrophysiology to investigate the role of BDNF signaling in mediating neuronal effects of mutations in MeCP2, the gene affected in Rett Syndrome. Although this is a well developed area of research, the authors provide a unique new set of insights showing that BDNF completely accounts for the observed cellular phenotypes and that the signaling is a local autocrine feed-forward mechanism that increases synapse number in a cell autonomous presynaptic fashion.

Essential revisions:

1) Two of the reviewers were concerned about the lack of a direct connection to in vivo analyses. This could be addressed by:a) Altering the text to limit the conclusions to those about the cell autonomous biology of isolated glutamatergic neurons from a Rett model, omitting those about the disease per se;b) Shoring up the in vivo observations to better integrate them into the rest of the manuscript. Specifically, nuclear size should also be measured in the cultured neurons and whether or not this is cell autonomous and BDNF-dependent should be assessed, ideally by comparing WT, mutant and BDNF overexpressing neurons. It would also be helpful to assess whether or not there are additional non-cell autonomous effects in vivo by comparing heterozygote to wild type animals.

2) Reviewers were also concerned about using two presynaptic markers rather than a pre- and postsynaptic marker to count synapses. At the very least, some of the observations should be repeated with an additional postsynaptic marker and the degree of correlation between pre-/post- identified synapses and pre-only synapses should be stated (for example Vglut and PSD95 would be good markers). In addition, it is important to rule out the possibility that synapse counts are influenced by the level of expression of the synaptic markers, as opposed to changes only in the physical presence or absence of synapses.

3) Reviewers felt that a clearer and more detailed explanation was needed concerning the basis for the authors' conclusion that there is no difference in BDNF release between WT and null cells, particularly given the NH_4_Cl data showing a lower fluorescence signal in the WT cells. In addition, it was felt that the authors need to clearly articulate a plausible hypothesis as to why autocrine, but not paracrine BDNF signaling would rescue the phenotypes they measured. It was suggested that characterizing TrkB phosphorylation under the conditions of bath application, and BDNF overexpression may help reveal the difference between autocrine and paracrine effects.

4) Reviewers were concerned that there was not sufficient evidence that BDNF overexpression was adequately achieved, especially in the mixed culture conditions. Also in the mixed cultures, how were contributions to synaptic measurements from excitatory synapses on interneurons excluded?

5) Was the person performing the analyses blind to the genotype/treatment of neurons?

---

## [Author Response]

*[…] Essential revisions:*

*1) Two of the reviewers were concerned about the lack of a direct connection to* in vivo *analyses. This could be addressed by:a) Altering the text to limit the conclusions to those about the cell autonomous biology of isolated glutamatergic neurons from a Rett model, omitting those about the disease per se;b) Shoring up the* in vivo *observations to better integrate them into the rest of the manuscript. Specifically, nuclear size should also be measured in the cultured neurons and whether or not this is cell autonomous and BDNF-dependent should be assessed, ideally by comparing WT, mutant and BDNF overexpressing neurons. It would also be helpful to assess whether or not there are additional non-cell autonomous effects* in vivo *by comparing heterozygote to wild type animals.*

As requested by the reviewers, we have addressed 1(b) by including the following data:

We have measured nucleus size in culture from both autaptic neurons (Figure 2—figure supplement 1) as well as co-cultured neurons (Figure 5). Please also see Results subsection “BDNF overexpression in *Mecp2^Null/y^* autaptic neurons restores evoked EPSC magnitude, RRP size, synapse number, dendrite length, soma and nucleus size” (for autaptic data) and “BDNF-induced TrkB activation regulates glutamatergic synapse number in a cell-autonomous and autocrine manner” (for co-culture data). In autaptic cultures, we found that nucleus size is smaller in *Mecp2^Null/y^* glutamatergic neurons and restored to WT levels by BDNF overexpression in a TrkB-dependent manner. In co-cultured neurons, we found that nucleus size remains reduced in *Mecp2^Null/y^* glutamatergic neurons although they are co-cultured with WT neurons. Furthermore, BDNF overexpression specifically in *Mecp2^Null/y^* neurons restored nucleus size up to WT levels. These findings demonstrate that change in nucleus size upon loss of MeCP2 is cell autonomous as well as BDNF-dependent in vitro. The analyses were done blind to genotype and treatment of neurons.

To further shore up the in vivo observations, we have additionally analyzed soma area of MeCP2+ and MeCP2- neurons in the hippocampal CA1 region of *Mecp2*^+/-^ mice (Figure 7). Please also see Results subsection “Neurons lacking MeCP2 have smaller somata and nuclei as compared to WT in Hippocampal CA1 of *Mecp2^+/-^* mice” and Materials and methods subsection “Quantification of neuronal somata and nuclei in hippocampal CA1”. We found that the soma of neurons lacking MeCP2 were 20% and 14% smaller than those of MeCP2+ neurons in 2- and 8-week old *Mecp2*^+/-^ mice, respectively. This phenotype is similar to that of nucleus size data in vivo. These data suggest that both soma area and nucleus size were reduced upon loss of MeCP2 in a cell autonomous manner in vitro and in vivo. Since both in vitro and in vivo data on nucleus and soma show significant reduction in MeCP2-deficient neurons in a mosaic condition, it is safe to conclude that cell autonomy (both in vitro and in vivo data) and BDNF dependence (from in vitro data) play a critical role in regulating neuronal growth and glutamatergic synapse formation. Because of the short time frame for revisions, we were only able to analyze soma area of MeCP2+ and MeCP2- neurons in *Mecp2*^+/-^ heterozygous mice and uncovered further cell autonomous effects. Nevertheless, we have discussed the role of putative non-cell autonomous effects in the sixth paragraph of the Discussion section.

*2) Reviewers were also concerned about using two presynaptic markers rather than a pre- and postsynaptic marker to count synapses. At the very least, some of the observations should be repeated with an additional postsynaptic marker and the degree of correlation between pre-/post- identified synapses and pre-only synapses should be stated (for example Vglut and PSD95 would be good markers). In addition, it is important to rule out the possibility that synapse counts are influenced by the level of expression of the synaptic markers, as opposed to changes only in the physical presence or absence of synapses.*

The reviewers raise an important point here: Are pre-/post- identified synapses and pre- only synapses correlated? This seems to be the case, as shown in the revised version of Figure 4 with new experiments. Please also see Results subsection “BDNF overexpression restores synapse density in *Mecp2^Null/y^* glutamatergic neurons”. We have labeled neurons with Homer1 as a postsynaptic marker, rather than PSD95 (for better specificity and signal to noise ratio in our hands), and VGLUT1 as a glutamatergic presynaptic marker; as seen in Figure 4. We then evaluated synapse density of both VGLUT1+ (Figure 4) and Homer1+ puncta (Figure 4) as well as VGLUT1-Homer1 colocalized puncta (Figure 4) and rate of colocalization of VGLUT1 to Homer1 (Figure 4). We found that both VGLUT1 and Homer1 synaptic densities were reduced to a similar degree in MeCP2-deficient neurons and rescued to WT levels upon both BDNF overexpression and exogenous BDNF application. To check specificity, we treated WT and BDNF overexpressing *Mecp2^Null/y^* glutamatergic neurons with the anti-neutralizing BDNF antibody and found that restored pre- and postsynaptic densities were reverted to *Mecp2^Null/y^* levels. We also measured VGLUT1-Homer1 colocalized puncta as well as rate of colocalization of VGLUT1 to Homer1 and found decrease in colocalized puncta upon loss of MeCP2, rescued by BDNF overexpression and exogenous BDNF in a BDNF-dependent fashion. In congruence, Chao 2007 analyzed colocalized VGLUT-PSD95 synaptic puncta as well as rate of colocalization in *Mecp2^Null/y^* neurons, and observed reduced VGLUT1-PSD95 puncta and reduced rate of colocalization, similar to our findings. Altogether, these data indicate maturation defect in excitatory synapses upon loss of MeCP2.

The reviewers also raise questions regarding the expression levels of synaptic markers used. To address this, we have measured expression levels of VGLUT1 in both autaptic and continental neuronal cultures, and Homer1 expression levels (see Figure 4—figure supplement 1). Please also see Results subsection “BDNF overexpression restores synapse density in *Mecp2^Null/y^* glutamatergic neurons”. We do not observe any changes in synaptic marker expression levels and so can conclude that expression levels of pre- or postsynaptic markers do not influence synapse counts.

*3) Reviewers felt that a clearer and more detailed explanation was needed concerning the basis for the authors' conclusion that there is no difference in BDNF release between WT and null cells, particularly given the NH_4_Cl data showing a lower fluorescence signal in the WT cells. In addition, it was felt that the authors need to clearly articulate a plausible hypothesis as to why autocrine, but not paracrine BDNF signaling would rescue the phenotypes they measured. It was suggested that characterizing TrkB phosphorylation under the conditions of bath application, and BDNF overexpression may help reveal the difference between autocrine and paracrine effects.*

We apologize for the lack of clarity in explaining the BDNF release data. BDNF release events were normalized to their corresponding peak NH_4_Cl events in WT and *Mecp2^Null/y^* neurons. This demonstrates that changes in NH_4_Cl data in WT and *Mecp2^Null/y^* neurons do not impact the quantification of BDNF release events. Hence, we can safely conclude that activity-dependent BDNF release remains unchanged upon loss of MeCP2. We have mentioned this in Materials and methods subsection “BDNF-superecliptic pHluorin live cell imaging” and in Figure 6—figure supplement 1.

Additionally, as asked by the reviewers, we have discussed in greater detail about autocrine versus paracrine BDNF signaling in the sixth paragraph of the Discussion section. While we agree with the reviewers that characterizing TrkB phosphorylation under different conditions is an interesting suggestion; we believe that autocrine BDNF signaling is predominantly responsible for rescue of measured phenotypes for the following reasons: (i) Our comprehensive analysis of morphological and physiological phenotypes in autaptic *Mecp2^Null/y^* neurons demonstrates that all measured phenotypes were rescued upon BDNF overexpression and exogenous BDNF application, in a TrkB-dependent manner. (ii) Specific restoration of glutamatergic synapse number and nucleus size in BDNF overexpressing *Mecp2^Null/y^* neurons co-cultured with WT neurons shows that these phenotypes are regulated in a cell autonomous manner via autocrine BDNF signaling. Furthermore, both phenotypes failed to be rescued by simply co-culturing WT neurons with *Mecp2^Null/y^* neurons, arguing against putative paracrine effects.

*4) Reviewers were concerned that there was not sufficient evidence that BDNF overexpression was adequately achieved, especially in the mixed culture conditions. Also in the mixed cultures, how were contributions to synaptic measurements from excitatory synapses on interneurons excluded?*

The reviewers raise an important point if BDNF expression was adequately achieved. This indeed seems to be the case as seen in Figure 2, where BDNF overexpression in autaptic *Mecp2^Null/y^* neurons convincingly rescued both morphological and physiological phenotypes. Furthermore, this rescue mimicked exogenous BDNF application data, as seen in Figure 3, in a BDNF- and TrkB-dependent manner as shown by BDNF neutralization and TrkB antagonist experimental data (Figure 2, Figure 3 and Figure 4). Additionally, commercial BDNF antibodies did not work effectively in our hands. Other elegant approaches to test specificity in future studies besides BDNF overexpression, neutralization and TrkB antagonist experiments, may include characterizing BDNF knockdown and rescue with exogenous BDNF. In the case of mixed co-cultures, we have strong proof of sufficient BDNF expression based on glutamatergic synapse number and nucleus size rescue data (Figure 5). These data also mimic the in vitro autaptic data (Figure 2) as well as in vivo soma and nucleus data (Figure 7). This co-culture data is particularly significant since we designed a unique in vitro experimental labeling technique to closely mimic Rett in vivo mosaicism, wherein we were able to locate synapse origin based on genotype. To our satisfaction, results matched autaptic and in vivo data, hence confirming robustness of the experimental design.

To address the second question (exclusion of excitatory synapses on to interneurons), we labeled all co-cultured WT and *Mecp2^Null/y^* neurons with VGLUT1, besides MAP2 and intrinsic Syp-mKate and Syp-GFP, to specifically mark excitatory synapses. We were not able to label additionally for the vesicular GABA transporter VGAT since all 4 channels were utilized for MAP2, VGLUT1, Syp-mKate and Syp-GFP. However, in all fields of view analyzed from three hippocampal cultures, we observed that the vast majority of Syp+ synapses were also VGLUT1+. Hence, we believe that excitatory synapses were primarily formed on to excitatory neurons, and contributions of synaptic measurements from excitatory synapses on to interneurons, would did not influence our findings.

*5) Was the person performing the analyses blind to the genotype/treatment of neurons?*

Yes, of three cultures analyzed, analyses for the first two cultures were done blind to both the genotype as well as treatment of neurons.